# Postnatal Protein Intake as a Determinant of Skeletal Muscle Structure and Function in Mice—A Pilot Study

**DOI:** 10.3390/ijms23158815

**Published:** 2022-08-08

**Authors:** Ifigeneia Giakoumaki, Natalie Pollock, Turki Aljuaid, Anthony J. Sannicandro, Moussira Alameddine, Euan Owen, Ioanna Myrtziou, Susan E. Ozanne, Ioannis Kanakis, Katarzyna Goljanek-Whysall, Aphrodite Vasilaki

**Affiliations:** 1Department of Musculoskeletal & Ageing Science, Institute of Life Course & Medical Sciences, Faculty of Health & Life Sciences, University of Liverpool, Liverpool L7 8TX, UK; 2The MRC—Versus Arthritis Centre for Integrated Research into Musculoskeletal Ageing (CIMA), University of Liverpool, Liverpool L7 8TX, UK; 3Department of Physiology, School of Medicine and REMEDI, CMNHS, NUI Galway, H91 TK33 Galway, Ireland; 4Department of Biotechnology, College of Science, Taif University, Taif 21944, Saudi Arabia; 5Chester Medical School, University of Chester, Bache Hall, Countess View, Chester CH2 1BR, UK; 6University of Cambridge MRC Metabolic Diseases Unit and Metabolic Research Laboratories, Institute of Metabolic Science, Addenbrooke’s Hospital Cambridge, Cambridge CB2 0QQ, UK

**Keywords:** maternal protein restriction, skeletal muscle, neuromuscular junction, offspring, microRNAs

## Abstract

Sarcopenia is characterised by an age-related decrease in the number of muscle fibres and additional weakening of the remaining fibres, resulting in a reduction in muscle mass and function. Many studies associate poor maternal nutrition during gestation and/or lactation with altered skeletal muscle homeostasis in the offspring and the development of sarcopenia. The aim of this study was to determine whether the musculoskeletal physiology in offspring born to mouse dams fed a low-protein diet during pregnancy was altered and whether any physiological changes could be modulated by the nutritional protein content in early postnatal stages. Thy1-YFP female mice were fed *ad libitum* on either a normal (20%) or a low-protein (5%) diet. Newborn pups were cross-fostered to different lactating dams (maintained on a 20% or 5% diet) to generate three groups analysed at weaning (21 days): Normal-to-Normal (NN), Normal-to-Low (NL) and Low-to-Normal (LN). Further offspring were maintained *ad libitum* on the same diet as during lactation until 12 weeks of age, creating another three groups (NNN, NLL, LNN). Mice on a low protein diet postnatally (NL, NLL) exhibited a significant reduction in body and muscle weight persisting up to 12 weeks, unlike mice on a low protein diet only prenatally (LN, LNN). Muscle fibre size was reduced in mice from the NL but not LN group, showing recovery at 12 weeks of age. Muscle force was reduced in NLL mice, concomitant with changes in the NMJ site and changes in atrophy-related and myosin genes. In addition, μCT scans of mouse tibiae at 12 weeks of age revealed changes in bone mass and morphology, resulting in a higher bone mass in the NLL group than the control NNN group. Finally, changes in the expression of miR-133 in the muscle of NLL mice suggest a regulatory role for this microRNA in muscle development in response to postnatal diet changes. Overall, this data shows that a low maternal protein diet and early postnatal life low-protein intake in mice can impact skeletal muscle physiology and function in early life while postnatal low protein diet favours bone integrity in adulthood.

## 1. Introduction

In humans, a 30–50% loss of muscle mass occurs between the ages of 50 and 80 years [1,2,3,4]. Muscle mass is dictated by the number and size of muscle fibres, and there is some evidence that the total number of muscle fibres in an individual muscle is established in utero or in early post-natal life, but the factors controlling both fibre number and initial muscle mass are unclear [5]. While all individuals lose muscle mass and develop age-related muscle weakness (termed sarcopenia), some individuals are more likely to reach clinically relevant levels that profoundly impacts on their quality of life, resulting in a reduced ability to carry out everyday tasks and increased susceptibility to falling. Furthermore, sarcopenia has been associated with various musculoskeletal disorders and poor prognosis in the context of various age-related outcomes [6,7].

Low birth weight, due to poor in utero nutrition, has been associated with subsequent reduced muscle mass and strength as the offspring reach older age, which may include a reduction in muscle fibre number [8]. Muscle strength in older subjects is reduced in individuals who did not grow well in early life [9], and it has been suggested that maternal, developmental and nutritional factors are important [10]. Meta analyses have revealed a positive association between birth weight and muscle strength, which is maintained across the life course. A key component of the maternal/early post-natal diet that is proposed to influence muscle development, mass and function is the protein content. Animal studies have demonstrated that pups born to mothers fed a low protein diet are significantly smaller [11,12] and that this sub-optimal maternal nutrition results in a reduction in muscle fibre size [13,14,15,16] and changes in the diameter of neuromuscular junctions (NMJs) [17].

In a similar manner, epidemiological studies in humans have shown that that there is a correlation between childhood growth and osteoporotic hip fracture risk in later life [18,19]. Early life nutritional effects on bone tissue have been also studied in animal models, where modulation of maternal diet during pregnancy and/or lactation impacts on skeletal homeostasis in the offspring [20,21]. It has been reported that adult offspring from protein restricted dams had lower bone mass in comparison with dams fed normal protein diets [22]. Thus, both in utero as well as early post-natal protein restriction appear to be associated with poor ageing of the musculoskeletal system and a shortened lifespan [8]. Studies have identified reduced muscle mass and function as well as altered bone development and bone cell activity throughout life that may impact the ageing process.

MicroRNAs (miRNAs) are small, single-stranded non-coding RNA molecules (20–22 nucleotides in length), that have the capacity to control gene expression mainly via direct binding to the 3′-untranslated region (UTR) region of their mRNA transcript [23,24]. Al-though miRNAs are known to control the expression of multiple transcripts in more than one tissue, expression of miRNAs may often be tissue specific with some miRNAs being more abundantly expressed in certain tissues [24]. MicroRNA-133 (miR-133) belongs to a conserved family of miRNAs known as the “myomiRs” and it is specifically and highly expressed in skeletal and cardiac muscles [25]. Previous studies have shown the involvement of miR-133 in skeletal muscle development [26] as well as neuromuscular interactions in vivo [27]. However, little is known about the role of miR-133 in skeletal muscle development following early-life malnutrition.

The aim of this study was therefore to determine the effects of reduced protein intake in utero and postnatally on NMJ, muscle structure and function as well as bone morphology during adulthood in mice. We hypothesised that maintaining pregnant dams on a low-protein diet will result in offspring with smaller muscle fibres and that optimisation of protein intake postnatally will result in increased muscle mass in the offspring leading to improved muscle function. The current work focused on 21 days (weaning) and 12 weeks of age (post sexual maturity) in order to identify potential early phenotypic events underlying the association between pre- and postnatal intake and muscle function and maintenance. Finally, we investigated whether changes in muscle and NMJ phenotype related to early life protein restriction are associated with changes in the levels of miRNAs. We identified miR-133 as a potential regulator of myogenesis, possibly through regulation of Atrogin-1, MuSK, FoxO3 and Sirt1 genes.

## 2. Results

### 2.1. The Effect of Dietary Pre- or Postnatal Protein Restriction on Body and Muscle Characteristics of 21-Days Old Mice

Mice born from dams maintained on a normal protein diet but fed postnatally by a foster dam maintained on a low-protein diet (NL) were visually smaller in body size and EDL muscle size, in comparison to mice fed a normal diet postnatally (NN and LN) (Figure 1A–C). As a consequence, mice from the NL group at 21 days of age demonstrated significant reductions in body weight and absolute muscle weight when compared with pups from the control (NN) group (Appendix A, respectively). Despite these differences, absolute muscle weight adjusted for total body weight was not significantly different between any of the groups (Appendix A). Microscopic evaluation of NMJs also showed no differences in muscle innervation (Appendix A); however, fibre size of EDL muscle, as assessed by Feret’s diameter, showed a significant decrease in the NL group (Figure 1D).

### 2.2. The Effect of Protein-Deficient Diet Pre- or -Postnatally on Muscle Weight, Total Body Weight and EDL Forces of 12-Week Old Mice

The body weights of the mice subjected to a protein-deficient diet postnatally (NLL) showed a significant reduction when compared to the body weights of mice from the control group (NNN). Prenatal protein restriction (LNN group) did not result in changes in body weight compared to the NNN group (Figure 2A). The absolute weights of EDL and TA muscles demonstrated no significant differences between mice from the three groups (Appendix A), whereas soleus (SOL), gastrocnemius (GTN) and quadriceps (QUAD) muscle weights of mice from the NLL group were significantly lower in comparison to the equivalent muscles of mice from the LNN and control (NNN) groups (Appendix A). However, absolute weights adjusted to total body weights were not significantly different between the three groups (Figure 2B; Appendix A).

Force generation of the EDL muscle of 12-week-old mice was recorded in situ, in order to assess whether potential physiological changes were associated with reduced muscle strength. EDL maximum force was significantly lower only in mice of the NLL group, compared to these of the control (NNN) group (Figure 2C). Specific muscle force showed no significant differences between the three groups of mice; however, a trend of lower specific force in NLL mice was observed (Figure 2D). Histological assessment of the structure of EDL muscle and analysis of fibre size revealed no significant differences between mice of the three groups (Figure 2E,F). Analysis of EDL fibre sizes showed similar distributions between mice of the three groups and no significant shift in distribution was recorded (Figure 2G).

### 2.3. Assessment of NMJ Structural Integrity

In order to assess whether reduction in EDL maximum force is attributed to impaired NMJ integrity and therefore a defective neuromuscular input, the structural integrity of the NMJ site was evaluated, with similar criteria as previously described [27]. Image analysis revealed the presence of morphologically abnormal NMJs in the EDL of mice from the NLL and LNN groups (Figure 3A). Examples of morphological abnormalities and denervation (partial or complete) of the AChR synaptic site are shown in Figure 3B.

Individual NMJs where scored into normal (N), morphologically abnormal (MA) and denervated (D) in mice of all three groups. Analysis of NMJs showed a significantly higher percentage of morphologically abnormal NMJs in mice of the NLL group, compared with those of the NNN (control) and LNN groups. The percentage of NMJs showing partial or full denervation was limited in mice from both NLL and LNN groups (Figure 3C). NMJs with morphological abnormalities were subsequently scored and subdivided into three categories: site fragmentation, size abnormalities or branching defects. The proportion of NMJs with abnormal size (small synaptic size), fragmented synaptic site or branching defects (limited branching) was significantly higher in mice of the NLL group, compared with mice from the control group (NNN) (Figure 3D–F).

### 2.4. Gene Expression Analysis of Marker Genes for Muscle Fibre Isoforms, Muscle Atrophy and NMJ Formation

To determine what molecular mechanisms may underlie reduced muscle and body size, as well as reduced muscle strength and NMJ abnormalities in mice on a low protein diet postnatally, expression analysis of genes involved in key mechanisms for muscle structure and function was performed. A significant increase in MyHC-IIa mRNA expression was observed in TA skeletal muscle of mice from the NLL group only as compared to the NNN control group (Figure 4B). Expression levels of MyHC-I (Figure 4A) and MyHC-IIb (Figure 4C) mRNA showed no significant differences between the three groups. Furthermore, expression analysis of Atrogin-1 gene revealed a significant increase in mRNA expression in mice of both the NLL and LNN groups, compared to the NNN group (Figure 4D). MuSK expression showed a trend towards higher levels in NLL mice; however, this was not significant (Figure 4E). No significant differences were observed in FoxO3 mRNA levels between mice of all groups (Figure 4F). Together, these data suggest that low protein diets early in life may result in changes in myosin type levels, indicating potential fibre type remodeling, as well as activation of muscle atrophy-related pathways.

### 2.5. The Effects of Protein Restriction on Bone during Adulthood

The assessment of trabecular bone parameters in the tibiae of 12 week–old adult mice revealed morphological changes. Bone mass, as reflected by bone volume to tissue volume percentage (%BV/TV), was found significantly higher in the NLL group as compared to NNN (23.41% increase) and LNN (25.15% increase) (Figure 5A,B respectively). In agreement with %BV/TV, the trabecular number (Tb.N) was significantly increased and trabecular separation (Tb.Sp) significantly reduced when NNN and NLL mice were compared (Figure 5D,E). However, there was no difference in trabecular thickness (Tb.Th) and, consequently, in trabecular pattern factor (Tb.Pf) and structural model index (SMI) parameters (Figure 5C,F,G), suggesting minor changes in the trabecular network microarchitecture. The respective comparisons between NNN and LNN groups showed no differences in either of the parameters. On the other hand, cortical thickness comparison in the tibial midshaft showed no differences between groups (Figure 5H,I), which underlies different effects on bone compartments.

### 2.6. Role of miR-133 in Skeletal Muscle In Vivo and In Vitro

Expression analysis of miR-133 revealed significantly lower levels in the TA muscle of NLL mice compared to NNN (Figure 6A), however, the levels in the sciatic nerve (SN) were very low and showed no differences between the three experimental groups (Figure 6B). This is consistent with miR-133 being enriched in skeletal muscle [27,28], while expression levels of other muscle-related miRs showed no differences except miR-128 (Appendix A).

In order to determine whether miR-133 could be a key regulator of skeletal muscle physiology of mice on a low protein diet postnatally, gain- and loss- of function experiments were performed in vitro, using C2C12 muscle cell cultures, and transfection efficacy was tracked fluorescently (Appendix A). Next, the effects of miR-133 overexpression or inhibition on cell viability were assessed using cytotoxicity assay. Analyses of cytotoxicity assay on C2C12 myoblasts treated with H_2_O_2_ following transfections revealed significantly lower cell death following overexpression of miR-133, compared to the Scr transfected cells (Figure 6C). Interestingly, both overexpression and inhibition of miR-133 had a positive effect on C2C12 myoblast number as the proliferation rate of C2C12 myoblasts significantly increased compared to the Scr control group as measured by proliferation assay (Figure 6D). This reflects the opposing literature on the role of miR-133 in regulating cell proliferation, which requires further analysis [25,29,30].

Fluorescent images of AM-133-transfected myotubes revealed a larger area occupied by myotubes per field of view, in comparison to the Scr control (Figure 6E). Quantification of this observation confirmed that AM-133-treated myotubes covered a larger area compared to NNN. Further to this, C2C12 myotubes transfected with AM-133 had a larger diameter compared to those transfected with miR-133 mimic (miR-133); however, miR-133-treated myotubes did not show smaller myotube diameter as compared to Scr controls, suggesting miR-133 may inhibit hypertrophy without promoting atrophy (Figure 6E,F). Notably, the fusion index of C2C12 myotubes transfected with either miR-133 or AM-133 revealed no significant difference compared to the control group, suggesting that miR-133 may control myotube growth independently of its role in myogenesis (Figure 6H).

### 2.7. miR-133 Regulates Expression of Myosins and Genes Associated with Hypertrophy Pathways

Given that miR-133 was downregulated in muscle of NLL mice and these mice displayed reduced muscle function and size (Figure 1 and Figure 2), the next logical step was to investigate whether the expression of the genes that were altered in the muscle of NLL mice was affected by miR-133 in C2C12 cells. Atrogin-1 expression was downregulated in C2C12 cells treated with AM-133 but also miR-133 (Figure 7A), suggesting this may be linked to miR-133 and AM-133 having a similar phenotypic effect on cell number (Figure 6D); however, this requires further exploration. Inhibition of miR-133 also led to increased levels of MuSK, FoxO3 and Sirt1 (Figure 7B–D), while MuSK also showed elevated, albeit not significantly, higher levels in muscle of NLL mice (Figure 4E). Muscle of NLL mice showed significantly higher levels of MyHC IIa expression, and this was mimicked in C2C12 cells treated with AM133 (Figure 7F). Moreover, AM-133 also led to increased levels of MyHC I expression, but not MyHC IIb (Figure 7E and Figure 7G, respectively), resulting in altered myosin percentages (Figure 7H). Together, these data suggest that downregulation of miR-133 in muscle of NLL mice may be associated with changes in the expression of different types of myosins and potentially fibre remodeling. Moreover, miR-133 may regulate expression of genes related to muscle hypertrophy and neuromuscular interactions.

## 3. Discussion

The aim of this study was to determine the effect of a reduced-protein diet pre- or postnatally on NMJ structure, muscle mass and function as well as on skeletal tissue in mice during early development and adulthood. Longitudinal assessment of body and muscle physiology and function in mice subjected to a protein-deficient diet was performed in order to investigate whether changes seen during gestation or lactation stages (21-day old mice) persist until early adulthood (12-week-old mice). Furthermore, this study aimed to assess at which developmental stage dietary modifications play a crucial role in skeletal muscle wasting.

Analysis of body weight from 21-day and 12-week-old mice and maximum force generation in EDL muscle at 12 weeks showed a significant decline only in mice on a low-protein diet at postnatal stages of development (NL for 21 days, NLL for 12 weeks). The reduction in body weight in 21-day and 12-week-old mice highlights the systemic effect of dietary interventions and the important role of nutrients during development. Changes in body weight following restriction of nutrients have been previously reported in both mice [12] and humans [31]. Several cases of malnutrition due to suboptimal protein intake have been recorded during infancy in humans even in developed countries, with a wide range of adverse effects [32]. Human studies have highlighted the link between low body weight early in life and reduced skeletal muscle function later in life [9]. In mice, early life protein restriction has been linked to lower body and organ weight along with altered expression levels of key proteins associated with muscle function and maintenance [12]. In this study, the weight of muscles composed of a mix of type I and type II muscle fibres (SOL, GTN and QUAD with SOL containing the highest % of type I fibres [33]) was significantly reduced following postnatal but not prenatal protein restriction (i.e., in NLL mice); however, this was not the case with the EDL and TA muscle, which are composed predominantly of type II muscle fibres. Muscle weight of mice from the LNN group showed no difference from control, but EDL muscle length was shorter in both NLL and LNN mice, compared to the control group (NNN). While specific muscle force showed no differences between the three groups, there was a significant reduction in maximum force generation in EDL muscle from NLL mice. These data indicate that changes in neuromuscular interactions, rather than muscle weight itself, could be associated with lower maximum force generation at that age. Absence of significant differences in mice in the LNN group suggests that pre-natal protein restriction does not have a direct impact on skeletal muscle weight and force generation during adulthood.

Histological analysis of the EDL muscle was necessary in order to assess any potential structural effect of maternal protein restriction in skeletal muscles of the offspring. At 21 days, EDL fibre size was significantly smaller in NLL mice only, compared to the NNN mice. However, this difference in myofibre size is lost in the EDL muscle at 12 weeks. Distribution analyses of the fibre size in EDL muscle showed no significant shift in the fibre size distribution. The lack of a significant reduction in the weight of the EDL muscles may be due to bulk gained via other sources such as fat infiltration, which was not investigated in this study.

Previous studies have demonstrated a strong link between muscle force and muscle physiology in terms of muscle fibre size. Weaker muscles have been previously associated with reduced number and muscle fibre size, one of the main characteristics of sarcopenia [34]. Recent studies have demonstrated a reduction in muscle strength during early ageing but in the absence of muscle atrophy, suggesting that reduction in the muscle force may be a predecessor of physiological changes seen in the muscle at later stages [35]. Although loss of muscle function without changes in muscle physiology has not been frequently reported, other factors may be equally responsible for a reduction in muscle strength. Examples of such factors include changes in MyHC isoforms, affecting the contractile properties of the muscle [36], innervation and signal transmission at the synaptic site of the muscle [37], changes in the levels of ROS [38], mitochondrial content [39] and changes in key molecular mechanisms regulating muscle mass and function [40].

In order to elucidate the mechanisms causing a reduction in maximum muscle force generation in 12-week-old mice on a protein-deficient diet postnatally, the NMJ structure was assessed. Firstly, NMJs were categorised into normal, morphologically abnormal or denervated (partial or complete) in order to examine whether changes in structure and morphology would be as severe as those seen in diseases [41,42,43] or during ageing [44]. Scoring of NMJs showed a significantly higher percentage of morphologically altered NMJs in mice of the NLL group compared to the NNN group, although perfect overlap between the pre-and post-synaptic sites was evident in the majority of NMJs. Furthermore, partial denervation was noted in a small portion of NMJs in both NLL and LNN mice groups, but their proportion was less than 5% of total NMJs scored and presented no significant differences compared to the control group (NNN). This small proportion of partially denervated NMJs was not surprising, considering that evidence of complete denervation in the muscle is not common in young/adult mice without a severe underlying condition such as muscular dystrophy, where more striking physiological effects are also noted [41]. It is not clear whether partial denervation can result in decreased muscle force, with data suggesting either [44,45].

Since significant differences in the NMJs with morphological abnormalities were recorded in the NLL group, subsequent scoring of those NMJs was performed. Three subcategories of morphological abnormalities were generated based on fragmentation of the overall synaptic site, limited or defective branching and small synaptic area. Similar criteria for NMJ scoring have been used in previous studies [46]. Data collected showed a significant increase in the proportion of the NMJs in all three subcategories in mice from the NLL group, compared to the control (NNN) mice. Unlike the NMJ phenotype observed in mice from the NLL group, no such changes in NMJ structure were observed in 12-week- old mice from the LNN group, which is in line with the normal force generation of the EDL muscle of these mice. Thus, early life protein restriction does not seem to contribute to aberrations of NMJ morphology during adulthood.

NMJ scoring indicate alterations in the morphology of the NMJ site but very limited evidence of partial denervation. Defects in the NMJs have been previously reported to impact skeletal muscle function; however, these are usually quite striking and often include partial or fully vacated AChR clusters [47]. These aberrations have been recorded in mice models with severe muscle defects [41] and during ageing [44]. It is possible that alterations in the morphology of the NMJs seen in this study may instead be the results of delayed development. Small NMJ size and limited branching in the synaptic site similar to that observed in mice from the NLL group have been recorded in several studies examining the developmental stages and maturation of NMJs in mice [48,49]. Assuming that mice from the NLL group exhibit delayed development, this would be in line with the small proportion of partially denervated NMJs and any site fragmentation seen, which could be due to NMJ remodelling rather than a deficit. Indeed, NMJ remodelling in mice during postnatal stages includes branch elimination, presence of unoccupied AChR clusters along with other structural changes [50,51]. Recent studies performing functional tests in muscle fibres occupied by an abnormal NMJ (e.g., fragmented) showed that such abnormalities per se do not affect the contractile properties of the skeletal muscle fibre [52]. However, there are studies showing a close association between NMJ morphology and muscle function in mice following injury [53]. Here, we advocate that the NMJ structural alterations observed in this study may be the result of underlying dysregulation in molecular mechanisms due to maternal protein restriction. This would result in a phenotype agreeing with delayed development in mice from the NLL group.

We evaluated the morphological effects of maternal protein diet on the skeletal tissue in 12-week-old mice as it corresponds to their peak bone mass. The evidence from a number of studies using animal models suggests that maternal undernutrition affects skeletal tissue homeostasis in the offspring during early life and in adulthood. In a study similar in design to this work, rats were maintained on 8% (low) or 16% (normal) maternal protein diets. Offspring at 8 weeks of age had significantly fewer colony-forming units fibroblastic (CFU-Fs) and alkaline phosphatase (ALP)-positive CFU-Fs, while ALP activity was reduced in the low-protein group. However, by 12 weeks, no significant difference was observed in the number of CFU-Fs but, importantly, ALP activity was significantly higher in the low-protein group [54]. Similarly, we found that trabecular bone mass in tibia was significantly elevated in the NLL group as compared to the control NNN. This suggests that the initial delay in bone development is then followed by a period of ‘catch-up’ growth [55]. Other studies using microCT reported that this effect is persistent in adulthood, resulting in changes in structural and mechanical properties of the offspring’s skeletal system with variation at different anatomical sites of the skeleton [56]. Further studies using older mice will unravel these effects in bone ageing.

To examine the hypothesis that protein restriction could result in altered molecular mechanisms in the muscle during development, expression of genes and miRNAs associated with muscle development and homeostasis were examined. miR-133 was downregulated in muscle of NLL mice compared to NNN mice (Figure 7). Expression analysis in TA muscle showed a significant upregulation of MyHC IIa mRNA levels in mice of the NLL group only and miR-133 inhibition in C2C12 cells led to higher levels of MyHC IIa; however, expression levels of MyHC I and -IIb showed no significant differences between the three groups. Increased intragroup variability in gene expression levels along with a low n number for each group were important restrictions for accurate estimation of the gene expression levels of these gene transcripts, and therefore a definitive conclusion could not be drawn.

Several studies have shown that during development the composition of skeletal muscle changes, with some MyHC isoforms being replaced. Specifically, EDL muscle is predominantly composed of MyHC-IIb fibres at the age of 21 days, when MyHC-IIa is still present. By 90 days of age, the EDL muscle is devoid of the type IIa isoform and is composed almost exclusively of type IIb fibres and, in a smaller proportion, type IId/x [57]. Very similar fibre type composition has also been observed in TA muscle, with MyHC-IIb being more abundant in adult mice [58]. Changes in muscle fibre composition, in terms of MyHC isoforms, could impact the contractile properties of skeletal muscles. Knockout mice for MyHC-IIb or -IId/x show distinct differences in muscle force generation [59]. Other studies have shown that adult mice lacking of MyHC-IId expression demonstrated increase expression of MyHC-IIa isoform, potentially acting as a compensatory mechanism [60]. Although MyHC-IIa and -IId/x fibres generate less force than -IIb fibres, upregulation of MyHC-IIa expression in mice from the NLL group might be a compensatory mechanism or an indicator of a developmental defect in their TA muscles. However, it is possible that this level of overexpression may not be sufficient to recover the muscle force in these mice. In order to examine whether this is indeed the case and whether these data are consistent in both EDL and TA muscles, immunofluorescent staining in transverse sections of those muscles would be necessary. Quantification of myofibres of different MyHC isoforms would provide an indication of potential shifts between different isoforms or the presence of hybrid myofibres in these muscles.

In addition to the expression patterns of MyHC isoform genes, differences in the expression levels of genes involved in muscle atrophy and synapse formation were also investigated. Atrogin-1 is a muscle-specific gene and it is highly expressed during skeletal muscle atrophy [61], although lower expression levels have also been reported in aged mice with sarcopenia [62]. Relative gene expression of Atrogin-1 in TA muscle showed a significant increase in the expression levels in mice from both the NLL and LNN group. Atrogin-1 levels were also altered in C2C12 cells following miR-133 level manipulation. Upregulation of this gene in mice of the NLL and LNN groups may be an early indicator of muscle fibre atrophy at later stages of adulthood. Given the main phenotype observed is changes in NMJ and muscle force, these data may indicate very early changes within the muscle associated with NMJ degeneration, which are currently not well understood. In terms of synaptic formation, expression levels of MuSK showed subtle changes in NLL mice; however, these changes need further validation. During postnatal development in mice, MuSK activation plays a fundamental role in NMJ maintenance and maturation in vivo [63,64]. Considering the structural changes seen in NMJ morphology in this study, it is likely that any changes in MuSK gene expression may not be evident yet. In order to assess the molecular changes underlying the morphological alterations seen in NMJs, it would be important to examine the expression levels of additional genes, including AChR-α, AChR-ε and AChR-γ subunit genes. Upregulation in gene expression of the AChR-γ gene may indicate possible damage at the NMJ site, including denervation [46]. The AChR-γ subunit is gradually substituted by the AChR-ε subunit during NMJ maturation at early stages of development [65]. Therefore, analysis of the expression level of the two AChR isoforms would be a useful tool to access whether delayed development or NMJ denervation/remodelling occurs in mice of the NLL group.

To assess whether the changes observed in vivo were attributed to downstream molecular mechanisms involved in skeletal muscle homeostasis, a key miRNA (miR-133) was investigated, using C2C12 muscle cells. Gain- and loss-of function experiments using miR-133 mimic and miR-133 inhibitor (AM-133) revealed a significant increase in C2C12 myotube diameter and total myotube area following inhibition of miR-133 (AM-133), although the opposite was not observed during overexpression of miR-133. Despite the increase in myotube area, no differences were observed in the fusion index in AM-133-transfected cells. These observations suggest overexpression of miR-133 does not cause atrophy but instead may inhibit hypertrophy. In contrast, inhibition of miR-133 may regulate myotube hypertrophy rather than myogenesis through extra fusion of cells, and hypertrophy of myotubes may be independent of cell fusion. This proposed mechanism is consistent with miR-133 regulation of FoxO3 and Sirt1 expression in C2C12 cells and is further supported by a study from McCarthy et al. (2011) [28] showing that satellite cells are not required for muscle fibre hypertrophy in mice. Previous studies have also shown that miR-133 expression is downregulated in skeletal muscle showing hypertrophy [66]. Inhibition of miR-133 in vitro caused no differences in cell death rate of C2C12 myoblasts and we speculate that inhibition of miR-133 could have triggered the endogenous cell overexpression of other miR-133 family members, providing protection of C2C12 myoblasts against cell death. This also supports the increase in the cell number observed in AM-133-treated C2C12 myoblasts, which could be the result of activation of different cellular mechanisms by other members of the miR-133 family. In vivo studies show that deletion of miR-133 does not cause changes in the skeletal muscle of mice until after 4 weeks of age, and this absence of phenotype could be attributed to the tissue specific expression of miR-133b [67]. Contrary to this, the rate of C2C12 cell death was significantly reduced after overexpression of miR-133, which is in line with the increased proliferation rate. Positive regulation of myoblast proliferation by miR-133 has been previously recorded in vitro [25]. Despite the data collected from in vitro and in vivo experiments in this study, it remains unclear whether miR-133 expression levels in skeletal muscle of mice is a direct consequence of the protein-deficient diet. Reduction in miR-133 levels may be the result of deregulation of molecular mechanisms directly affected by the protein-deficient diet, which could differ depending on the developmental stage in which this diet was introduced. As such, the mechanisms affecting miR133 expression or the mechanisms affected by the downregulation of miR-133 expression could differ between the NLL and LNN groups of mice. Transcriptomic analysis and identification of predicted target genes as direct targets of miR-133 would provide a clearer view into the exact molecular mechanisms affected by changes in the expression levels of miR-133 in mice following a protein-deficient diet pre-or postnatally.

## 4. Materials and Methods

### 4.1. Animals and Experimental Groups

All experimental work involving animals was performed under appropriate project licenses (PPL 40/3620 and 70/8378) and inspected by the UK Home Office in accordance with guidelines under the UK Animals (Scientific Procedures) Act 1986. Animal use followed the 3Rs guidelines. Mice were kept at the Biomedical Services Unit (BSU) of the University of Liverpool and monitored daily for any health and welfare issues. Breeding pairs were originally purchased from The Jackson Laboratory (The Jackson Laboratory, Bar Harbor, ME, USA; Thy-1 YFP-16, Stock# 003709). Mice were bred from homozygous breeding pairs and were fed a standard laboratory diet *ad libitum*. All mice were maintained under barriers and were exposed to a 12 h dark, 12 h light cycle.

C57BL/6 Thy1-YFP16 transgenic mice express yellow fluorescent protein (YFP) with high specificity in the motor and sensory neurons at high levels (with no expression in non-neuronal cells and no apparent toxic effects). Neuron-specific expression of YFP allows the visualisation of motor neurons and muscle innervation from mid-gestational stages, providing a reliable approach in the assessment of structural alterations in motor neurons and NMJs without the use of antibody staining [44].

Two weeks before mating, nulliparous female and age-matched male mice were fed either the low-protein diet (5% Crude Protein W/W ISO’ (P), Code 829202; Special Diet Services, Essex, UK) or a control diet (20% Crude Protein W/W ISO’ (P), Code 829206, Special Diet Services, Essex, UK). All newborn pups were cross-fostered within 24 h after birth to different lactating dams maintained on either a 20% or a 5% protein diet. In the first set of experiments, pups were culled at weaning age (21 days) and categorized as: Normal-to-Normal (NN), Normal-to-Low (NL) and Low-to-Normal (LN); n = 6 in each group (Figure 1A). We also attempted to use an LL (Low-to-Low) group, however the number of pups born did not reach n = 6 and this group was abandoned. Further groups of mice were weaned at 21 days old onto either the low-protein (5% protein) or the 20% normal protein diet to produce the 3 following groups: NNN, NLL and LNN (n = 5–15 per group), depending on the available litter, and were maintained on those diets for 12 weeks. Following sacrifice by cervical dislocation, body weight and the weight of various hindlimb muscles were immediately recorded and prepared as described below. Some of the mice were used for force measurements, and others for tissue dissection and fibre size or RNA analyses.

Limitations: Following weaning, mice were fed ad lib. We were not able to measure the exact food intake for each mouse individually as mice were housed in groups. However, based on our observations any differences in food intake were subtle.

### 4.2. EDL Force Measurement

Extensor digitorum longus (EDL) muscle has a high percentage of fast-twitch muscle fibres (MyHC type II isoforms) [68], which are preferentially lost during ageing. Therefore, this muscle is ideal for measurement of the contractile properties of skeletal muscle in mice. EDL muscle force via nerve stimulation was recorded in 12-week-old mice. Under terminal anaesthesia (via intraperitoneal (IP) injection 66 mg/kg ketamine hydrochloride, 0.55 mg/kg medatomidine hydrochloride), the distal tendon of the EDL muscle was exposed and secured to the lever arm of a servomotor (Cambridge Technology, Cambridge, UK). The knee of the hindlimb was fixed and bipolar platinum wire electrodes were placed across the exposed peroneal nerve. Optimal length (Lo) of the EDL was recorded using serial increments in muscle length at 1 Hz stimulation and finally set at the length that generated the maximal force (Lo). The Po of the EDL was determined following electrical stimulation of the muscle in order to contract at Lo with optimal stimulation voltage (8–10 V) every 2 min for 300 ms with a 0.2 ms pulse width. The frequency of stimulation was increased from 10 to 50 Hz and followed by 50 Hz increments to a maximum of 300 Hz. When the maximum force of the muscle reached a plateau, despite the increase of the stimulation frequency, the Po was recorded.

### 4.3. Histological Analyses

For analysis of muscle structure, EDL and tibialis anterior (TA) muscles were coated with OCT (Cell Path, Newtown, UK) and snap-frozen in liquid nitrogen-cooled isopentane (Fisher Scientific, Loughborough, UK). Transverse sections of the EDL muscles (12 μm thickness) cut using a Leica 1890 cryotome (Leica Biosystems, Newcastle upon Tyne, UK) were collected on SuperfrostPlus glass slides (ThermoScientific, Horsham, UK) and allowed to dry for 1 h at room temperature. Muscle sections were fixed with ice-cold methanol (Sigma-Aldrich, Gillingham, UK) for 10 min followed by two washes with 0.04% Tween-20 in phosphate buffered saline solution (PBS; Sigma-Aldrich) for 5 min each. For identification of the extracellular matrix of the muscle fibres, EDL and TA sections were stained with 1:1000 fluorescein wheat germ agglutinin (WGA; 5 μg/mL; Vector Laboratories Ltd., Peterborough, UK) for 10 min, followed by two washes with 0.04% Tween-20 in PBS solution (Sigma-Aldrich) for 5 min each. Hard-set with DAPI (Vector Laboratories Ltd.) was used as a nuclear stain and mounting medium. Staining was conducted at room temperature at all times. Images were acquired using a C1 Nikon Eclipse Ti confocal laser scanning microscope (Nikon, Tokyo, Japan) at 20× magnification. All fibres from 3 sections per mouse were analysed. Image analysis was performed using ImageJ (NIH) software. All analyses were performed in a semi-automated fashion, using the “Tissue Cell Geometry Stats” macro (Institute for Research in Biomedicine, University of Barcelona, Spain; http://adm.irbbarcelona.org/image-j-fiji#TOC-Automated-Multicellular-Tissue-Analysis, assessed on 30 March 2018). Muscle fibre size was determined using the minimum Feret’s diameter.

For NMJ analysis, the EDL muscles were fixed in 10% neutral-buffered formalin (10% NBF; Sigma-Aldrich) for 1 h, followed by two rinses with PBS solution. Tissues were then permeabilised with 1% Triton X-100 in PBS solution for 30 min. For visualisation of the motor end plate, EDL muscles were stained with α-bungarotoxin Alexa Fluor™ 594 conjugate (B13423; 1:500 dilution; Molecular Probes, Life Technologies Ltd., Renfrew, UK) for 30 min, washed with 0.04% Tween-20 in PBS solution for 20 min and transferred to 0.1% PBS-NaN_3_ solution until imaged. Staining was conducted at room temperature at all times. Images of NMJs were acquired using a Nikon Eclipse Ni-E intravital confocal microscope at 60× magnification. NMJs (50–250) from each sample were scored and separated into three main groups: Normal (N), morphologically altered (MA) and denervated (D) (both partially and completely vacant endplates were included). Morphologically altered NMJs were divided into three subcategories: overall synaptic site size, lack of extensive branching and “non-pretzel” or fragmented morphology of the overall synaptic site.

### 4.4. Cell Culture, Transfection, Cytotoxicity Assay and Immunostaining

C2C12 myoblast cells (ATCC, New York, NY, USA) were cultured in Dulbecco’s modified Eagle’s medium (DMEM) (Sigma-Aldrich) growth medium containing 10% fetal bovine serum (FBS) (Sigma-Aldrich), 1% Glutamax (Gibco, Loughborough, Leicestershire, UK) and 1% penicillin-streptomycin (Sigma-Aldrich) at 37 °C. Transfections of C2C12 cells were performed as previously described [69]. Briefly, C2C12 myoblasts were seeded at a density of 50,000 cells/well in laminin-coated (1 mg/mL; Sigma-Aldrich) 6-well plates and transfected with either 100 nM of miR-133 mimic, 100 nM of miR-133 inhibitor (antagomir, AM) or 100 nM of scrambled (Scr) negative control (Appendix A). The differentiation medium, containing 2% horse serum (Sigma-Aldrich) as a replacement of FBS in the growth medium, was added 6 h after transfection. The transfection protocol was repeated on day 3 and 5 post-seeding to enhance efficiency of transfection in myotubes. Immunostaining with MF20 was performed 7–10 days after transfection using DAPI to visualize cell nuclei as previously described [69] and images were captured using the Nikon eclipse Ti-E inverted confocal microscope. For cytotoxicity assay, cells were first treated with 100 nM of miR-133 mimic or antagomir (AM133) or scrambled control. Six hours after transfections, cells were treated with 10 μM H_2_O_2_ and incubated for 18 h. Cytotoxicity assay was performed according to the manufacturer’s protocol (Cytotox 96, Promega, Madison, WI, USA). Cell proliferation was measured using the CCK-8 proliferation kit (Sigma) following transfections.

### 4.5. Sample Preparation and RNA Extraction

RNA isolation was performed from left-over TA muscle tissue where available, sciatic nerve (SN) and C2C12 cells. For muscle tissue, samples were ground in liquid nitrogen using a pestle and mortar, while for cell samples, the wells were washed twice with pre-warmed Dulbecco’s PBS, total RNA was isolated using TRIzol (Invitrogen, Renfrewshire, UK). RNA concentration and purity were estimated according to the 260/280 and 260/230 ratio recorded using Nanodrop2000 (ThermoFisher).

### 4.6. cDNA Synthesis and RT-qPCR

First strand cDNA synthesis for miRNA and mRNA was performed using 100–200 ng RNA from TA muscle, the entire SN and cell samples using the T100 Thermocycler (BioRAD, Watford, Hertfordshire, UK), using previously described methods [70]. Briefly, mRNA cDNA synthesis was performed using SuperScript™ IV VILO™ Master Mix (Invitrogen) according to the manufacturer’s protocol. For miRNAs, miScript II RT Kit (Qiagen, Manchester, UK) was used and cDNA synthesis was performed according to the manufacturer’s instructions.

Real-time quantitative PCR (RT-qPCR) was performed using the BioRad CFX Connect™ Real-Time PCR Detection System in 20 μL reaction volume. Primer sequences (Appendix A) were designed to span an exon-exon junction and produce a 50–70 nt amplicon. Gene expression relative to 18S for mRNA and Snord-61 for miRs (Appendix A) was calculated using the ΔΔCt method. The qPCR conditions were: 95 °C for 10 s, 58 °C for 30 s, 72 °C for 30 s for mRNA and 95 °C for 30 s, 55 °C for 30 s and 72 °C for 30 s for miRNAs (40 cycles) using a hot start step of 95 °C for 15 s. Samples in which a gene expression was not amplified were not included in the analysis.

### 4.7. Micro-Computed Tomography (microCT)

Following skeletal muscle dissection, the hindlimb bones were harvested, fixed in 10% NBF solution for 24 h, extensively washed with PBS and scanned with microCT using a Skyscan 1272 scanner (Bruker, Kontich, Belgium; 0.5 Al filter, 50 kV, 200 mA, voxel size 4.5 μm, 0.3° rotation angle step). Reconstruction of the image datasets was performed using NRecon and regions of interest were selected using Dataviewer and CTan software. Trabecular and cortical parameters were analysed using CTAn in the proximal metaphysis and midshaft using 400 and 100 slices, respectively [71,72]. For trabecular bone analysis, mineralised cartilage served as a reference point [73].

### 4.8. Statistical Analysis

All datasets were statistically analysed with GraphPad Prism 6 software and expressed as the mean ± standard deviation (mean ± SD). Statistical comparisons were performed using one-way ANOVA with Dunnett’s post-hoc analysis, using NN or NNN as the control group. A *p* value of less than 0.05 was considered statistically significant.

## 5. Conclusions

In conclusion, dietary protein restriction appears to have a more detrimental effect during postnatal stages of development and changes persist until early adulthood. Such changes include a reduction of EDL muscle force, which is evident only in mice on a protein-deficient diet postnatally and might be the combined result of defective NMJs and changes in the molecular machinery controlling the skeletal muscle phenotype. On the contrary, prenatal protein restriction followed by normal protein intake during postnatal stages of development may result in molecular alterations without evidence of any substantial phenotypical changes. Early life changes within skeletal muscle and neuromuscular interactions may be partially regulated via miR-133 through its regulation of genes associated with fibre types and hypertrophy. In addition, postnatal protein deprivation seems to have a positive effect on bone accrual in adulthood; however, the possible mechanisms, such as nutrition-induced microRNA changes or other epigenetic modifications, as well as the long-term impact, need further investigation.

## Figures and Tables

**Figure 1 ijms-23-08815-f001:**
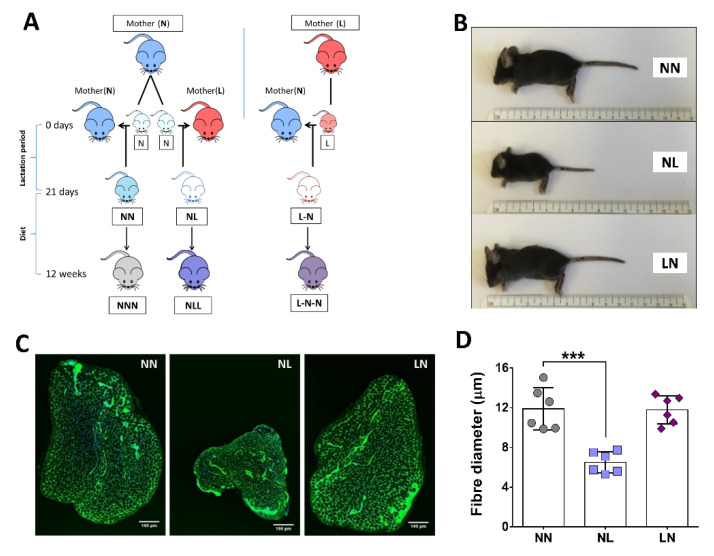
The effects of reduced protein intake in utero or postnatally on 21-day old mice. Representation of the experimental design (**A**). NL mice were smaller in body size (**B**) and had smaller EDL muscle (**C**) with significantly reduced muscle fibre size (**D**). NN was used as the control group for all statistical comparisons. All data are presented as mean ± SD. N = 6; *** *p* < 0.001; scale bar: 100 μm.

**Figure 2 ijms-23-08815-f002:**
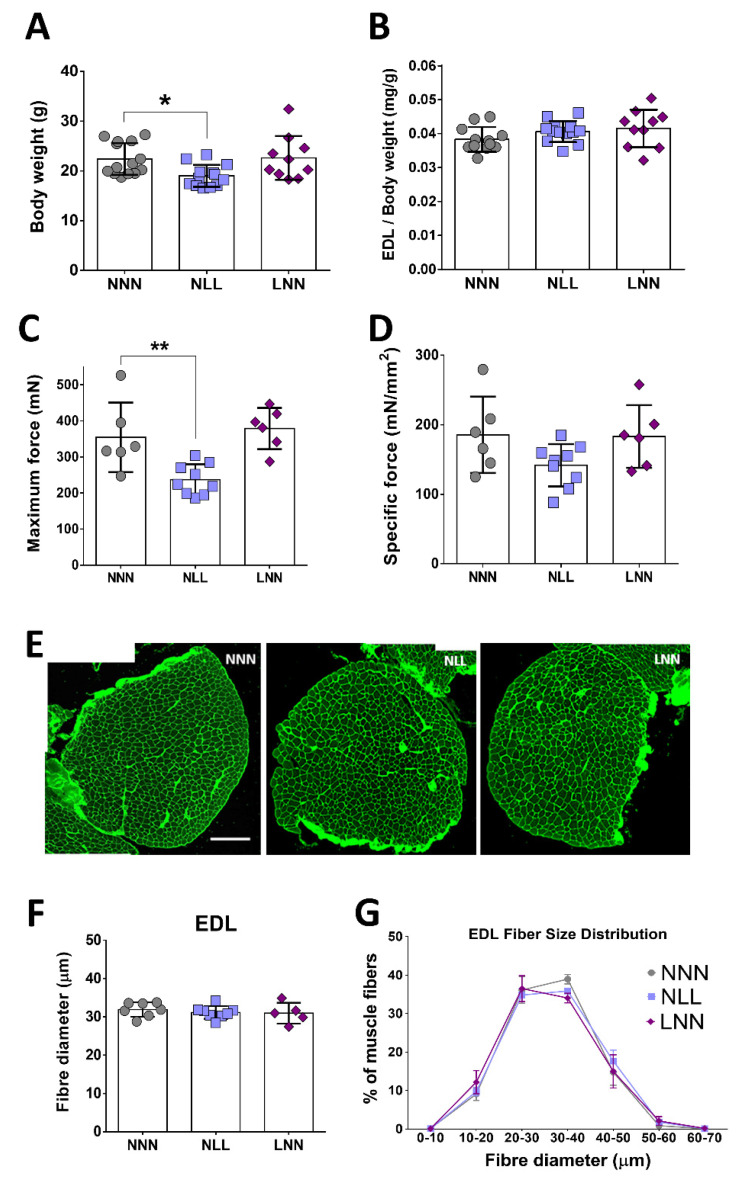
The effects of reduced protein intake in utero or postnatally on the body and muscle weight as well as EDL muscle forces of 12–week–old mice. NLL mice showed reduced total body weight (**A**), but no difference was observed when absolute muscle weights of EDL were adjusted to total body weights (**B**). Maximum force showed a reduction in the NLL group of mice (**C**), while analysis of the specific tetanic force revealed no differences (**D**). Representative images of histological examination of the EDL muscle of 12-week-old mice are shown (**E**). EDL fibre size, expressed as Ferret’s diameter, showed no differences between groups (**F**). Myofibre size distribution analysis of EDL muscle, expressed as a percentage of total muscle fibres analysed in EDL muscle of 12-week-old mice, revealed no changes between groups (**G**). NNN was used as the control group for all statistical comparisons. Scale bars = 200μm, * *p* ≤ 0.05 and ** *p* ≤ 0.01 (mean ± SD; n = 5–14).

**Figure 3 ijms-23-08815-f003:**
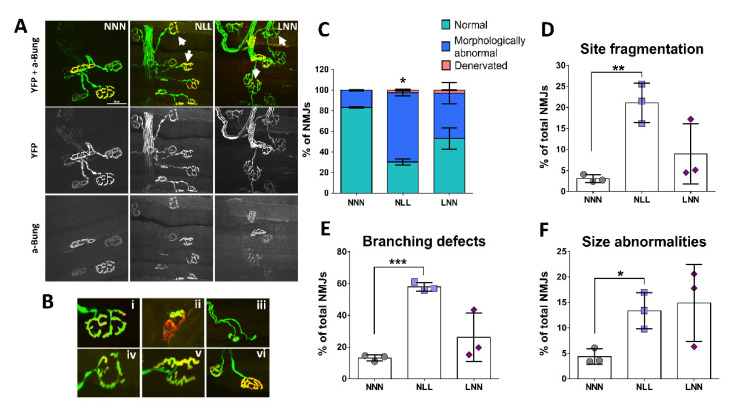
Fluorescent images of NMJ sites in EDL muscle of 12-week-old mice. The pre-synaptic terminal expressing YFP (green) and the post-synaptic end plate stained with α-bungarotoxin (red) are in full alignment. Representative images of NMJ sites in mice of NLL and LNN groups show morphological aberrations (arrow), in comparison with the “pretzel” shape seen in the control group (**A**). Example images of: (**B**) Normal “pretzel shape” NMJ site (i), partially denervated NMJ site, where AChR site is vacated (ii), NMJ with no perforation (iii), limited NMJ branching (iv), fragmented NMJ site with no “pretzel-like” shape (v) and small size of NMJ site (vi). Classification of structural changes of the NMJ site in EDL muscle of 12-week-old mice showed a high proportion of morphological abnormalities (**C**) such as increased site fragmentation (**D**), with limited/defective branching (**E**) or with small size of the synaptic area (**F**) only in the NLL group as compared to NNN. * *p* ≤ 0.05, ** *p* ≤ 0.01, and *** *p* ≤ 0.001 (mean ± SD; n = 3 animals in each group, minimum 80 and maximum 250 NMJs were assessed per each muscle. Scale bar: 50 μm; magnification: 60×.

**Figure 4 ijms-23-08815-f004:**
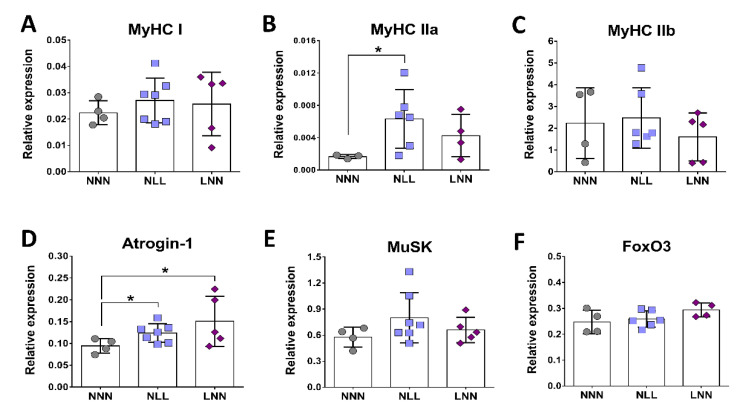
Expression analysis of marker genes for MyHC composition, muscle atrophy and NMJ formation in TA muscle of 12-week-old mice. Mice born from dams fed a normal protein diet and maintained on the same diet (NNN) were used as controls in comparison with mice born from dams fed a normal protein diet during gestation, switched to low protein intake during lactation by cross-fostering and maintained on the low protein diet after weaning (NLL) and mice fed a low protein diet in utero and maintained on normal protein post-weaning until adulthood (LNN). Relative gene expression levels of MyHC-I (**A**), MyHC-IIa (**B**), MyHC-IIb (**C**), Atrogin-1 (**D**), MuSK (**E**) and FoxO3 (**F**). * *p* ≤ 0.05 (mean ± SD; n = 3–7).

**Figure 5 ijms-23-08815-f005:**
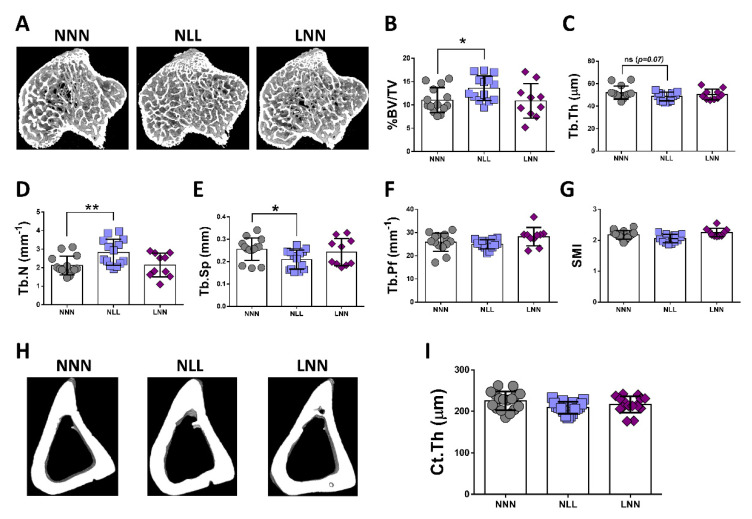
Micro-computed tomography morphological measurements in tibiae of 12-week-old mice. Representative images of the NNN, NLL and LNN groups (**A**) and quantification of bone volume to tissue volume ratio (%BV/TV) (**B**), trabecular thickness (Tb.Th) (**C**), trabecular number (Tb.N) (**D**), trabecular separation (Tb.Sp) (**E**), trabecular pattern factor (Tb.Pf) (**F**) and structural model index (**G**). For cortical bone (**H**), the cortical thickness (**I**) was measured and compared, where NNN served as the control group for all statistical comparisons; * *p* ≤ 0.05 and ** *p* ≤ 0.01, (mean ± SD; n = 10–14).

**Figure 6 ijms-23-08815-f006:**
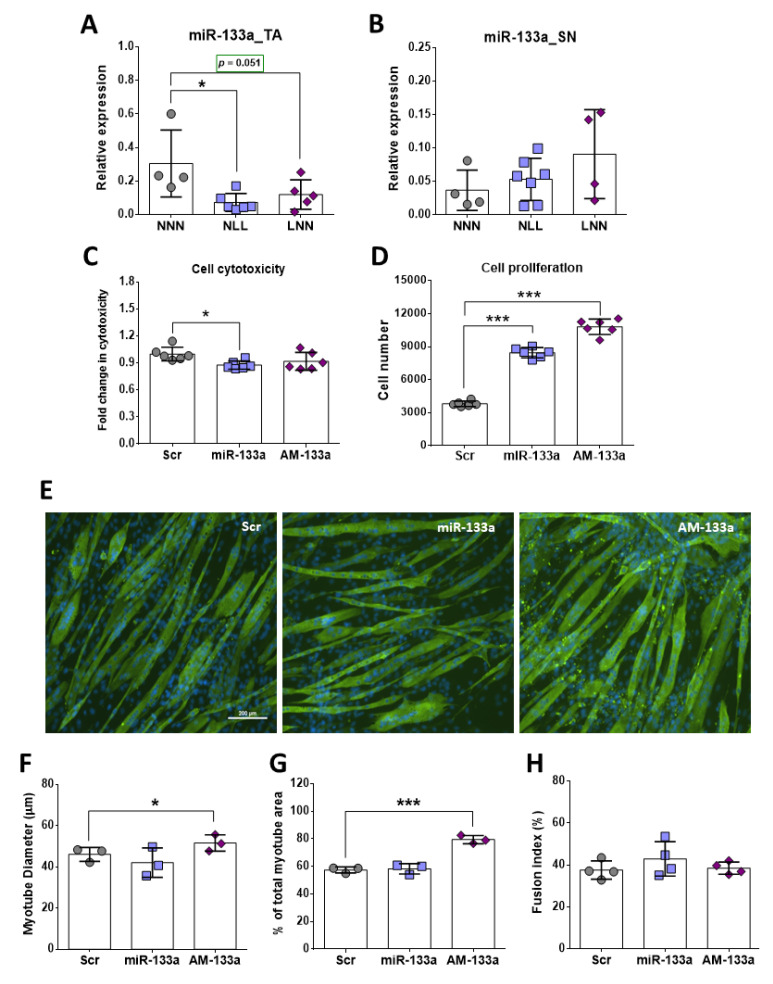
Expression analysis of miR-133 in TA muscle (**A**) and sciatic nerve (**B**). CytoTox96 cytotoxicity assay (following treatment of cells with H_2_O_2_ and miR-133/AM133) (**C**) and CCK-8 proliferation assay (**D**) of C2C12 myoblasts following transfection with Scr, miR-133 or AM-133. Representative images of C2C12 myotubes transfected with miR-133 or AM-133 stained for myosin heavy chain (MyHC) and nuclei (**E**) (MF20 = Green; DAPI = Blue). Myotube diameter (**F**) area per field of view (**G**) and fusion index (**H**) were used as measures of myotubes. Scale bars = 200 μm. * *p* ≤ 0.05 and *** *p* ≤ 0.001 (mean ± SD; n = 3–7).

**Figure 7 ijms-23-08815-f007:**
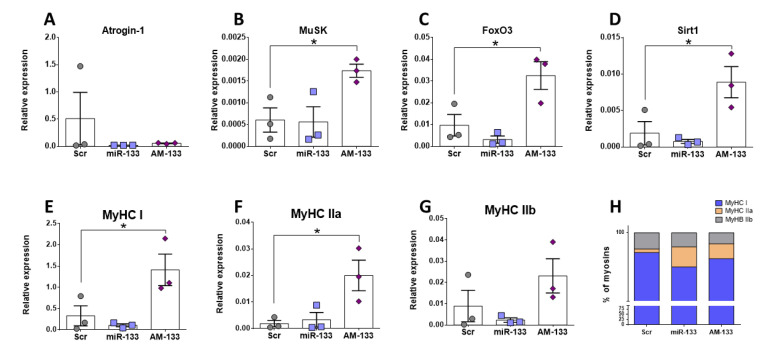
Expression (relative to 18S) analysis of marker genes for MyHC composition, atrophy and NMJ formation in C2C12 cells treated with Scr control, miR-133 mimic or anatgomiR-133 (AM-133). Gene expression was analysed for Atrogin-1 (**A**), MuSK (**B**), FoxO3 (**C**), Sirt1 (**D**), MyHC I (**E**), MyHC IIa (**F**), MyHC IIb (**G**). Myosin percentages are shown in (**H**). * *p* ≤ 0.05 (mean ± SD; n = 3).

## Data Availability

Not applicable.

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
