# Peer review of "Postnatal Protein Intake as a Determinant of Skeletal Muscle Structure and Function in Mice—A Pilot Study"

_ijms, 2022, doi:10.3390/ijms23158815_

Round 1

Reviewer 1 Report

This study was to determine whether the musculoskeletal physiology in offspring born to mouse dams fed a low-protein diet during pregnancy was altered and whether any physiological changes could be modulated by the nutritional protein content in early postnatal stages. Therefore, it is expected to provide important information for suggesting the importance of nutrition in the early postnatal stages for effective muscle growth, and is well-written overall.

Data on bone development is included in the results, and it is described in the abstract, but in the conclusion, there is no description of this part at all. Why? I think a little extra skill is needed.

Author Response

Reviewer #1:

This study was to determine whether the musculoskeletal physiology in offspring born to mouse dams fed a low-protein diet during pregnancy was altered and whether any physiological changes could be modulated by the nutritional protein content in early postnatal stages. Therefore, it is expected to provide important information for suggesting the importance of nutrition in the early postnatal stages for effective muscle growth and is well-written overall.

We thank the reviewer for these encouraging comments.

Data on bone development is included in the results, and it is described in the abstract, but in the conclusion, there is no description of this part at all. Why? I think a little extra skill is needed.

Many thanks for this comment. Obviously, this was omitted during manuscript preparation and drafts’ editing between co-authors. This part of the conclusions for protein diet effects on bone tissue has now been added in lines 647-650.

Reviewer 2 Report

The study is well written. The pictures are arranged in an aesthetic manner. The flow has been clearly defined. The topic is clinically relevant. The article deserves publication. There are some minor suggestions.

First, the following reference can be added for emphasize the association between sarcopenia and bad health consequences:

https://pubmed.ncbi.nlm.nih.gov/34026779/

https://pubmed.ncbi.nlm.nih.gov/35752855/

Second, the full term of “SD” should be given when it first appears in the article.

Third, in Figure 4, please provide the full term of NNN, NLL and LMN.

Author Response

Reviewer #2:

The study is well written. The pictures are arranged in an aesthetic manner. The flow has been clearly defined. The topic is clinically relevant. The article deserves publication. There are some minor suggestions.

Many thanks for these kind comments.

First, the following reference can be added for emphasize the association between sarcopenia and bad health consequences:

https://pubmed.ncbi.nlm.nih.gov/34026779/

https://pubmed.ncbi.nlm.nih.gov/35752855/

Many thanks for these suggestions. The citations have been now added in lines 53-55.

Second, the full term of “SD” should be given when it first appears in the article.

Many thanks for this observation. The abbreviation is now added in line 239.

Third, in Figure 4, please provide the full term of NNN, NLL and LMN.

Although these terms are explained in detail in Methods, the full terms are now added to the legend to Figure 4.